# Identification of gene function based on models capturing natural variability of *Arabidopsis thaliana* lipid metabolism

Sandra Correa Córdoba [1,2] ✉, Hao Tong [1,2], Asdrúbal Burgos[3], Feng Zhu[4], Saleh Alseekh [5,6], Alisdair R. Fernie [5,6] & Zoran Nikoloski [1,2,6] ✉

Lipids play fundamental roles in regulating agronomically important traits. Advances in plant lipid metabolism have until recently largely been based on reductionist approaches, although modulation of its components can have system-wide effects. However, existing models of plant lipid metabolism provide lumped representations, hindering detailed study of component modulation. Here, we present the Plant Lipid Module (PLM) which provides a mechanistic description of lipid metabolism in the *Arabidopsis thaliana* rosette. We demonstrate that the PLM can be readily integrated in models of *A. thaliana* Col-0 metabolism, yielding accurate predictions (83%) of single lethal knock-outs and 75% concordance between measured transcript and predicted flux changes under extended darkness. Genome-wide associations with fluxes obtained by integrating the PLM in diel condition- and accession-specific models identify up to 65 candidate genes modulating *A. thaliana* lipid metabolism. Using mutant lines, we validate up to 40% of the candidates, paving the way for identification of metabolic gene function based on models capturing natural variability in metabolism.

Lipids comprise a class of structurally diverse molecules that facilitate compartmentalization of biochemical activities and are involved in critical cellular functions[1–3]. The structural diversity of plant lipids is generated by an underlying metabolic network whose fine-tuned regulation adjusts the lipidome to internal and exogenous cues[4], relevant in agricultural and biotechnological applications[5,6]. A considerable part of our knowledge concerning the regulation of plant lipid metabolism has been obtained through reductionist approaches[7,8]. In contrast, approaches from the constraint-based modeling framework[9] can be used with genome-scale models (GEMs), that include mechanistic descriptions of lipid metabolism, to: (i) characterize how lipid metabolism is interconnected with pathways of primary metabolism and (ii) design manipulation strategies for lipid-related traits[10,11].

In comparison to the GEMs of yeast[12–14] and microalgae[15–17], however, that include a comprehensive description of lipid metabolism, with few notable exceptions[18–21], most existing plants metabolic models do not consider reactions compartmentalization, pathway redundancy, and enzyme promiscuity in lipid metabolism. This results in a largely lumped representation of lipid metabolic pathways[5] in existing plant models, reflecting the limited degree of annotation of plant genomes. Hence, despite numerous data sources[22–25] and pathways databases[26–28] the use of automated tools to reconstruct lipid metabolic pathways is not warranted even for

[1]Bioinformatics, Institute of Biochemistry and Biology, University of Potsdam, Potsdam, Germany. [2]Systems Biology and Mathematical Modelling, Max Planck Institute of Molecular Plant Physiology, Potsdam, Germany. [3]Department of Zoology and Botany, University of Guadalajara, Guadalajara, Mexico. [4]National R&D Center for Citrus Preservation, Hubei Hongshan Laboratory, National Key Laboratory for Germplasm Innovation and Utilization for Horticultural Crops, Huazhong Agricultural University, Wuhan, China. [5]Central Metabolism, Max Planck Institute of Molecular Plant Physiology, Potsdam, Germany. [6]Center of Plant Systems Biology and Biotechnology, Plovdiv 4000, Bulgaria. ✉e-mail: Cordoba@mpimp-golm.mpg.de; nikoloski@mpimp-golm.mpg.de

the model plant *Arabidopsis thaliana* (Arabidopsis). This in turn limits the applications of constraint-based modeling to gain better understanding of plant lipid metabolism.

To capitalize from the constraint-based modeling framework, here we provide a reconstruction of a Plant Lipid Module (PLM), including a mechanistic description of lipid metabolism in the Arabidopsis rosette. The PLM can be readily integrated in a semi-automated fashion into a metabolic model of any size (Fig. 1). We demonstrate the functionality of the resulting tool and the accuracy of predictions by integrating the PLM into five plant metabolic models and through three case studies in which we predict and validate: (i) the effects of gene lethality on growth, (ii) the response of lipid metabolism to extended darkness, and (iii) putative candidates of genes that modulate leaf lipid metabolism under extended darkness. The validated genes involved in plant lipid metabolism are identified based on flux genome-wide association (fGWA) performed with predicted fluxes from Arabidopsis accession-specific metabolic models that include the PLM. Therefore, predictions from the PLM provide model-driven insights in the natural variability of lipid metabolism across Arabidopsis accessions, demonstrating its utility as a complement to experimental genome-wide association studies.

## Results
### Generation and integration of the PLM into plant metabolic models

The reconstructed PLM comprises 5956 reactions and 3108 metabolites in 16 compartments (Fig. 1a). It includes metabolic pathways for the biosynthesis and degradation of structural, storage and signaling lipids, grouped into 26 lipid classes (Supplementary Data 1). In addition, it comprises the pathways necessary for the generation of essential organic cofactors and metabolic intermediates (Supplementary Method 1).

The PLM can be integrated into existing metabolic models via an easy-to-use tool (Fig. 1b). To this end, we tested the integration of the PLM in five published plant metabolic models of different size[19,21,29–31] that: (i) have COBRA model structure[32]; (ii) include metabolites with molecular neutral and/or charged formulas, accompanied by the respective charges and identifiers (e.g. KEGG and/or ChEBI); and (iii) can be imported and exported with standard software packages/toolboxes (e.g. via COBRA Toolbox)[33]. The integration of the PLM was completed efficiently even for the largest model (Supplementary Table 1). In the following, we presented the results of the expanded model based on the medium-scale AraCore model[29] as a template.

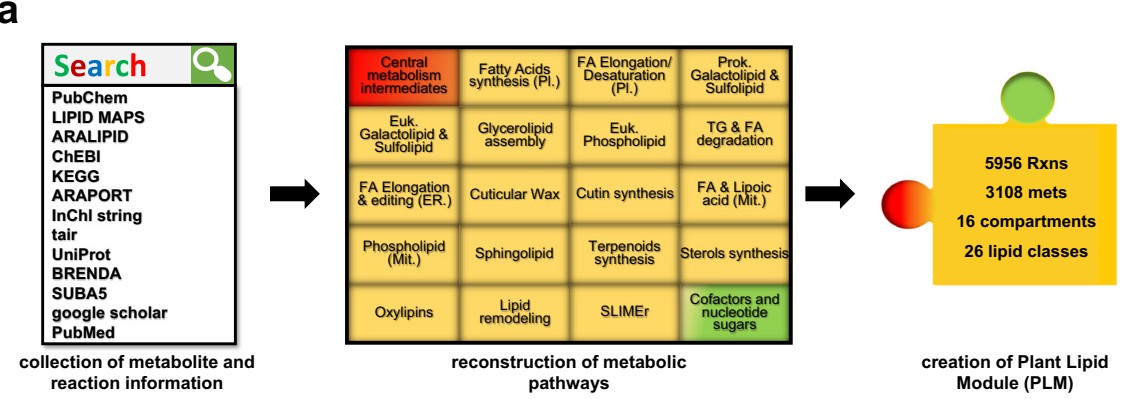

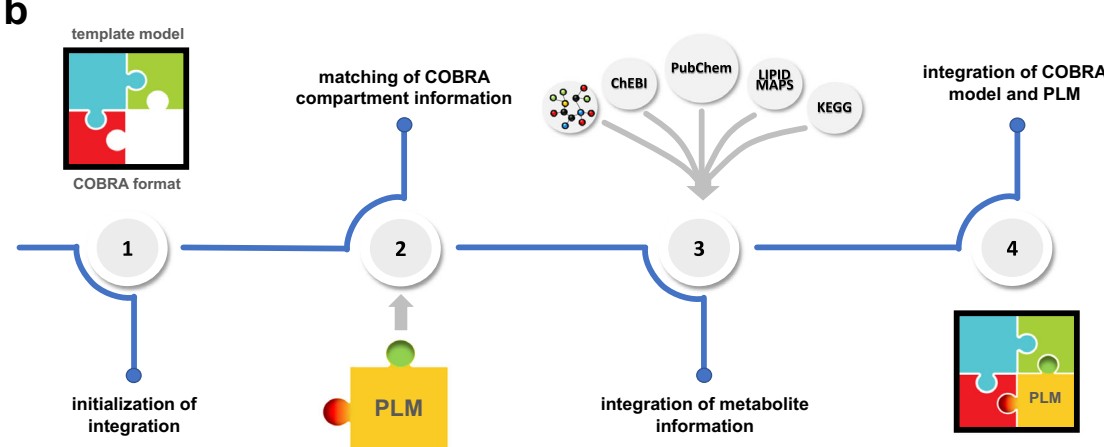

**Fig. 1 | Diagram of the steps for the integration of the Plant Lipid Module (PLM) into a metabolic model. a** The PLM comprises the existing up-to-date information from heterogeneous data sources, including the metabolism of structural, storage and signaling lipids, grouped into 26 lipid classes, in addition to the set of essential organic cofactors and metabolic intermediates. **b** The integration of the PLM into a template metabolic model comprises four steps: (1) selecting the template model; (2) consolidating the information of the compartments in the template model and the PLM; (3) consolidation of the information on the metabolites shared between the PLM and the template model and adding to the latter the metabolites only present in the PLM; (4) adding the PLM reactions to the template model, followed by the removal of duplicated reactions. COBRA constraint-based reconstruction and analysis, ER endoplasmic reticulum; euk. eukaryotic, FA fatty acid, mets metabolites, mit. mitochondria, pl plastid, prok. prokaryotic, rxns reactions, SLIMEr split lipids into measurable entities reactions[14], TG triacylglycerol.

## PLM improves predictions of lethal genes

The functionality of the PLM was initially assessed through a synthetic lethality analysis that was performed before and after its integration into the template model. Our results demonstrated that the incorporation of the PLM resulted in a larger number of synthetic lethal genes (up to 47% more) compared to the template model itself (Supplementary Table 2). The accuracy of the classification of the genes according to the synthetic lethality analysis was verified by using the available experimental evidence from Arabidopsis mutants[27,34] (see Supplementary Data 2 and 3). From 627 genes in the template model before the integration of the PLM, 108 were identified as single lethal genes; of these, for thirteen there was experimental evidence from mutants in the SeedGenes Project website[34], of which lethality was confirmed for 10 genes (Fig. 2a). In contrast, after the integration of the PLM in the template model, the number of genes increased to 1091; 174 of these genes were found to be single lethal, of which 29 were in line with experimental evidence from SeedGenes Project[34] (Fig. 2b). Furthermore, for an additional 85 genes from the extended model evidence was obtained from the ARALIPID website[27]. Therefore, our results showed that up to 86% of the lethality predictions were aligned with the experimentally observed phenotype (Fig. 2c). In contrast, for the template model before the integration of the PLM it was possible to find evidence of lethality only for two genes. In addition, we found more false-positive predictions in the template model before the integration of the PLM (up to 23%), in comparison to the expanded model (up to 17%). Therefore, our results demonstrate that the integration of the PLM resulted in better accuracy (83-88%) (Fig. 2b, c), compared to that of the template model (77-80%) (Fig. 2a). In addition, we found that genes involved in double synthetic lethals are part of metabolic pathways that present a higher degree of robustness given the existence of isoenzymes and/or pathway redundancy. These pathways included the production of lipids, carbohydrates, nucleotide sugars, adenosine 5′-triphosphate (ATP), and sugar phosphates (see Supplementary Fig. 1 and Supplementary Table 3).

## PLM predicts reliable condition- and accession-specific flux distributions

The integration of the PLM in accession-specific metabolic models provides the basis for investigating the natural variability and genetic architecture of fluxes in lipid metabolism. To this end, we created a diel model[35] that includes the PLM to simulate fluxes for extended darkness, a condition known to induce changes in the lipidome of vegetative tissues[36–41]. The resulting model was rendered accession- and condition-specific by creating biomass reactions that incorporate experimental data measured in a population comprised of 284 Arabidopsis natural accessions belonging to the HapMap panel[42], that were exposed to the condition of extended darkness[43].

We used these accession- and condition-specific models with parsimonious flux balance analysis (pFBA)[44] to calculate flux distributions for each accession. We found that the predicted flux distributions were in line with evidence available for the condition-dependent (de) activation of reactions for different pathways of lipid metabolism. For instance, we predicted: (i) the activity of chloroplast FAs synthesis and FAs elongation pathways in dark (Fig. 3a), but with reaction rates substantially reduced compared to the flux values predicted in the light[45–47] (Fig. 3b); (ii) enrichment of phosphatidylcholine (PC) with polyunsaturated FAs, through the acyl editing cycle[48,49], that despite having low reaction rates (Fig. 3b), was active in all accessions (Fig. 3a, Supplementary Data 4); (iii) reduced activity in biosynthetic pathways of various classes of phosphoglycerolipids (PGLs) (e.g. phosphatidylglycerol (PG), phosphatidylinositol (PI), and phosphatidylethanolamine (PE)) (Fig. 3a, b); besides a strong decline in the reaction rates of PC synthesis (Fig. 3b) in agreement with the reduced levels measured for the above PGLs[41]; (iv) higher activity displayed by reactions involved in chloroplast membrane lipid degradation mostly through

hydrolysis (Fig. 3a, lipid remodeling−DGL), together with the reduced proportion of active reactions for the eukaryotic and prokaryotic galactolipid synthesis pathways (Fig. 3a, b), consistent with the observed breakdown of major membrane lipids of chloroplast (e.g. monogalactosyldiacylglycerol (MGDG) and digalactosyldiacylglycerol (DGDG)[41]; (v) reduced activity of the de novo synthesis pathways of several precursors of neutral lipids (e.g. triacylglycerol−TG) such as lysophosphatidic acid (LPA), phosphatidic acid (PA) and diacylglycerol (DG) (Fig. 3a), in parallel to the enhanced activity of the pathways of neutral lipids synthesis. The latter agrees with the reported accumulation of TG in the ER[48] (Fig. 3a, b, TG synthesis−ER) and chloroplasts[38,48,50] (Fig. 3a, b, lipid remodeling−PES) that occur at the expense of membrane lipid degradation upon recycling the resultant FAs and DG; and (vi) the increased flux rates of reactions responsible for TG hydrolysis (Fig. 3b), and the enhanced FA degradation via β-oxidation (Fig. 3a, b), in line with the observed greater activity of neutral lipid degradation pathways[41]. This is also accompanied by the induced transcription of specific genes assigned to FAs peroxisomal β-oxidation[51–53], and genes coding for isoforms of the lipase enzyme[53,54]. Interestingly, we also predicted that the degradation routes were inactive in 15% and 0.4% of the accessions after 3 and 6 days of extended darkness, respectively (see Supplementary Data 4); that deviates from the experimental evidence for plants exposed to extended darkness[41]. A detailed analysis of the flux distributions allowed us to identify that the predicted maltose consumption rates were higher in the accessions with inactive degradation routes (see Supplementary Fig. 2).

We also extended the lethal gene analysis to identify if the results were accession- and condition-specific. We found that the knockout of genes from the single lethal set resulted in a consistent prediction of non-viable phenotypes for all accessions (Supplementary Data 5). On the other hand, for the double lethals, we identified that 10.2% of gene knockout combinations for which the predicted phenotype was not lethal in some of the accessions under the considered conditions (Supplementary Data 6). The latter was attributed to the accession-specific biomass composition since the remaining model reactions are invariant across the models used.

## Concordance of transcriptomic and fluxomic changes in Col-0

Since natural selection acts towards improving resource use efficiency, changes in expression of genes are expected to entail changes in corresponding fluxes. Therefore, we next determined the correspondence between the patterns of activation/deactivation of reactions with the increase/decrease expression of the respective genes for Arabidopsis Col-0 exposed to extended darkness. In line with our expectations, we found that the pathways with reduced reaction activity for FA elongation[51,54] the synthesis of FAs[51,53,54], galactolipids[51,53,54], and PGLs[41] (e.g. PE, PC, and PG) (Supplementary Data 7), and the de novo synthesis pathways for LPA, PA, and DG (see Supplementary Data 7, GPAT, LPAAT and PAP reactions) included genes with decreased expression. In addition, we found increased expression of genes related to pathways for which enhanced activity was found, including: TG synthesis in the ER and chloroplasts (via PES enzyme), TG hydrolysis, β-oxidation, and chloroplast membrane lipids degradation (via DGL enzyme) (Supplementary Data 7). However, we also identified discrepancies to our expectation, such as the increased expression of genes associated with PI synthesis, although the pathway was predicted to have a low activity[41] (Supplementary Data 7); similarly, the genes coding for a phospholipase (NPC5) and a cholinephosphotransferase (PDCT) showed no significant changes in the transcript levels (see Supplementary Data 7), although the reactions catalyzed by the gene products were predicted to be active (Fig. 3a, lipid remodeling−NPC5 and phospholipid metabolism).

To quantify the correspondence between flux and gene expression changes, we used the gene-protein-reaction (GPR) rules. The latter

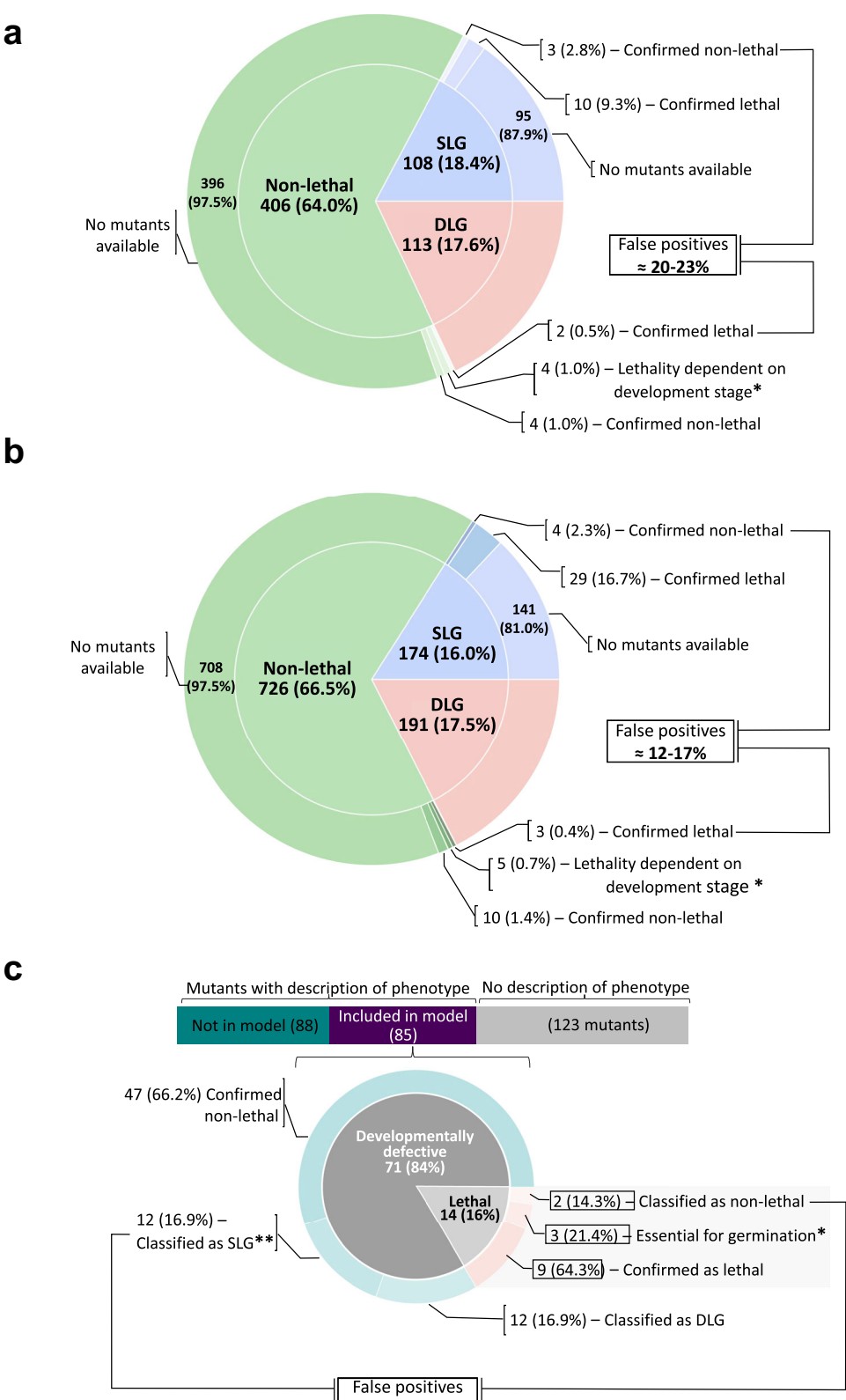

are represented as logical rules that describe the relationships between genes, their protein products and the reactions they catalyze[55]. Thus the rules capture how gene products concur to catalyze the associated reaction, and therefore served to map gene expression from two transcriptomics data sets measured in Arabidopsis Col-0 exposed to extended darkness[53,54]. We found that the direction of changes between fluxes and expression levels agreed in up to 76% of the enzymes for which data were available (referred as concordant enzymes). The remaining enzymes (up to 25%), corresponded to the so-called outliers, in which there was no agreement in the direction of change (Table 1, Supplementary Figs. 3 and 4). These results were consistent between the two transcriptomic data sets ($\chi^2(1, N = 472) = 0.05$; $p = 0.831$; effect size $= 0.01$; chi-square test; Table 1).

**Fig. 2 | Synthetic lethal gene analysis.** Genes of the AraCore model before and after the integration of the PLM were classified as single- (SLG), double- (DLG) and non-lethal genes according to results of the lethality analysis. **a** The lethal sets obtained for the AraCore model before the integration of the PLM were validated using evidence from 25 characterized Arabidopsis mutants available in SeedGenes Project database[34]. The information from ARALIPID website[27] was not used in this case because it contained data from mutants for only two genes included in the AraCore model. The validation was then made for the lethal sets identified for the AraCore model expanded with the PLM. We used (**b**) data for 60 mutants from SeedGenes and (**c**) information for 85 mutants from ARALIPID. The accuracy of the results was verified by comparing the SLG and non-lethal sets with the phenotype exhibited by the respective mutant. The cases in which the phenotype did not correspond to the lethality classification were considered as false positives. The DLG set was not used for the validation since the lethality is given by the simultaneous deletion of a pair of genes, which makes it difficult to find characterized Arabidopsis mutant plants with the double gene knock-outs. Source data are provided as a Source Data file. *Evidence obtained from SeedGenes and ARA-LIPID points that the mutation of these genes is embryo lethal. Conversely, the lethality analysis predicted the deletion of these genes as a non-lethal mutation. The discrepancy arises from the fact that the model was built for leaves, where in addition to these genes, high levels of expression have also been reported for genes that encode isoenzymes. Therefore, the essentiality of these genes depends on the developmental stage. **The genes encode lipid-related enzymes identified as SLG, but the corresponding phenotypes only exhibits developmental defects. Seven of the twelve genes are related to plastidial transport of lipid species and to cutin and wax synthesis pathways, which are not fully elucidated. Discrepancies may result from the existence of isoenzymes not yet annotated, or alternative routes not elucidated to date.

We hypothesized that the presence of outlier reactions, whose flux and corresponding transcript expression change did not agree, may be due to post-translational modifications (PTMs). To test the hypotheses we employed the FAT-PTM database[56] to obtain a list of the PTMs reported to date for the enzymes (Fig. 4) which we used in a comparative analysis based on the classification of concordant (Supplementary Data 8) and outlier (Supplementary Data 9) enzymes. Contrary to our expectations, we did not identify significant differences between the type and proportion of the PTMs that are present in the outlier and concordant enzymes (Supplementary Data 10). Therefore, other types of regulation (e.g. allostery) may explain the discrepancy between the changes in transcript and fluxes.

## Candidate genes modulating lipid metabolism identified by flux GWA

The predicted condition- and accession-specific flux distributions were in turn used to determine the genetic architecture of lipid metabolism fluxes in Arabidopsis leaves. To this end, we determined genome-wide association for each of the predicted fluxes in every condition using the available single-nucleotide polymorphisms (SNPs) (Supplementary Table 4). We found that a modest number (up to 5%) of fluxes were associated with SNPs in enzyme-encoding genes included in the GPR rules of the underlying reactions. Further, a higher number of fluxes (up to 25%) were associated with SNPs in genes that encode enzymes catalyzing reaction steps immediately upstream or downstream of the corresponding reaction flux, corroborating the relevance of the findings. In addition, for up to 70% of fluxes, the significant SNPs occurred in genes encoding transcription factors (TFs) (Fig. 5, Supplementary Table 5). Two TFs, *WRKY18* (AT4G31800) and *TREE1* (AT4G35610), were identified as candidate genes in fluxes predicted for control (light) and extended darkness conditions (Fig. 5, Supplementary Note 1 and Supplementary Figs. 5–8). *WRKY18* is known to be expressed in leaves, where it positively modulates the expression of defense-related genes and disease resistance[57]. Besides, it could also play a potential role in cuticle formation, since it is among the most up-regulated and highly expressed genes in top stem epidermis[58]. *TREE1* is primarily expressed in flowers[59], and the available experimental evidence points to its role in the regulation of transcriptional repression of shoot growth in response to ethylene[60].

Upon further examination of the TF-coding genes as putative regulators of fluxes, we found that between 53–64% contained potential DNA-binding domains for the genes in the GPR rules of the corresponding reactions (Supplementary Table 5). In further support of our findings, between 82–86% of the TF-coding candidates related to lipid metabolism are expressed in leaves (Supplementary Note 1). We identified *WRINKLED 1* (*WRI1*–AT3G54320) (Supplementary Note 1, Supplementary Figs. 5 and 6), a regulator of embryo development and maturation, that also regulates the expression of several enzymes of the glycolytic and FA synthesis pathways[5]. In addition, the TFs found associated with the largest number of fluxes (>32) are related to

various biological processes, for instance, embryogenesis and organogenesis, hormone signaling, DNA transcription, senescence, response to abiotic factors (light) and development phase transition (Supplementary Data 11).

For the remaining fluxes, we found that up to 37% were associated with candidates for which it was not possible to establish the type of protein encoded. Of these, up to a fifth of the underlying reactions do not have GPR rules assigned, of which the majority (up to 92%) are related to transport of metabolites (Supplementary Table 5). Although the allocation of the transport reactions in the PLM reconstruction was based on extensive literature search (see Supplementary Method 1), it is also true that a good part of the intra-/extra-cellular transport processes remain to be fully characterized.

To validate the putative candidates regulating fluxes in lipid metabolism, we used the lipid profiles for a collection of 364 Arabidopsis T-DNA insertion lines[61] (Supplementary Data 12 and 13). From 60 enzyme-coding gene candidates identified by fGWA across conditions, we found that up to 19% of the corresponding T-DNA lines resulted in changes in the relative abundances of the lipids involved in the underlying reactions (Supplementary Table 4, Supplementary Data 14). For instance, these gene candidates are involved in acyl editing, transglycosylation, glycerophospholipid metabolism, transport of PGLs and PA, and synthesis of PA, DG, and TG. On the other hand, we found that in up to 68% of TF-coding candidates, there is: (i) information about the potential association (e.g. DNA-binding domains) among the knocked-out enzymatic genes and the TFs, and (ii) have potential binding sites with genes associated to reactions that produce lipids which exhibited changes in relative abundance in mutants (Supplementary Table 4, Supplementary Data 15). Validation of the remaining candidates would necessitate finer lipidomic characterization of the mutants for which currently data from broader lipid analysis were not available[61]. Together, the validation demonstrated that up to 40% of the putative candidate genes predicted to modulate lipid metabolism could be validated against published data sets from mutant lines (Supplementary Table 4), paving the way for expanding such model-driven fGWA to other pathways.

## Discussion

Through the creation of the PLM, we made a comprehensive compilation of the biochemical knowledge of lipid metabolism in vegetative tissues (for which there is sufficient evidence to date) for the model plant Arabidopsis. The functionality of the PLM was demonstrated through several case studies. For instance, we simulated the influence of environmental conditions (i.e. extended darkness) on the modulation of lipid metabolism and related pathways, where overall, we found a good agreement with the available experimental evidence. We also carried out gene essentiality analysis, where the integration of the PLM allowed the identification of a much larger number of lethal genes compared to the original model. Despite these advances, our results pointed at a

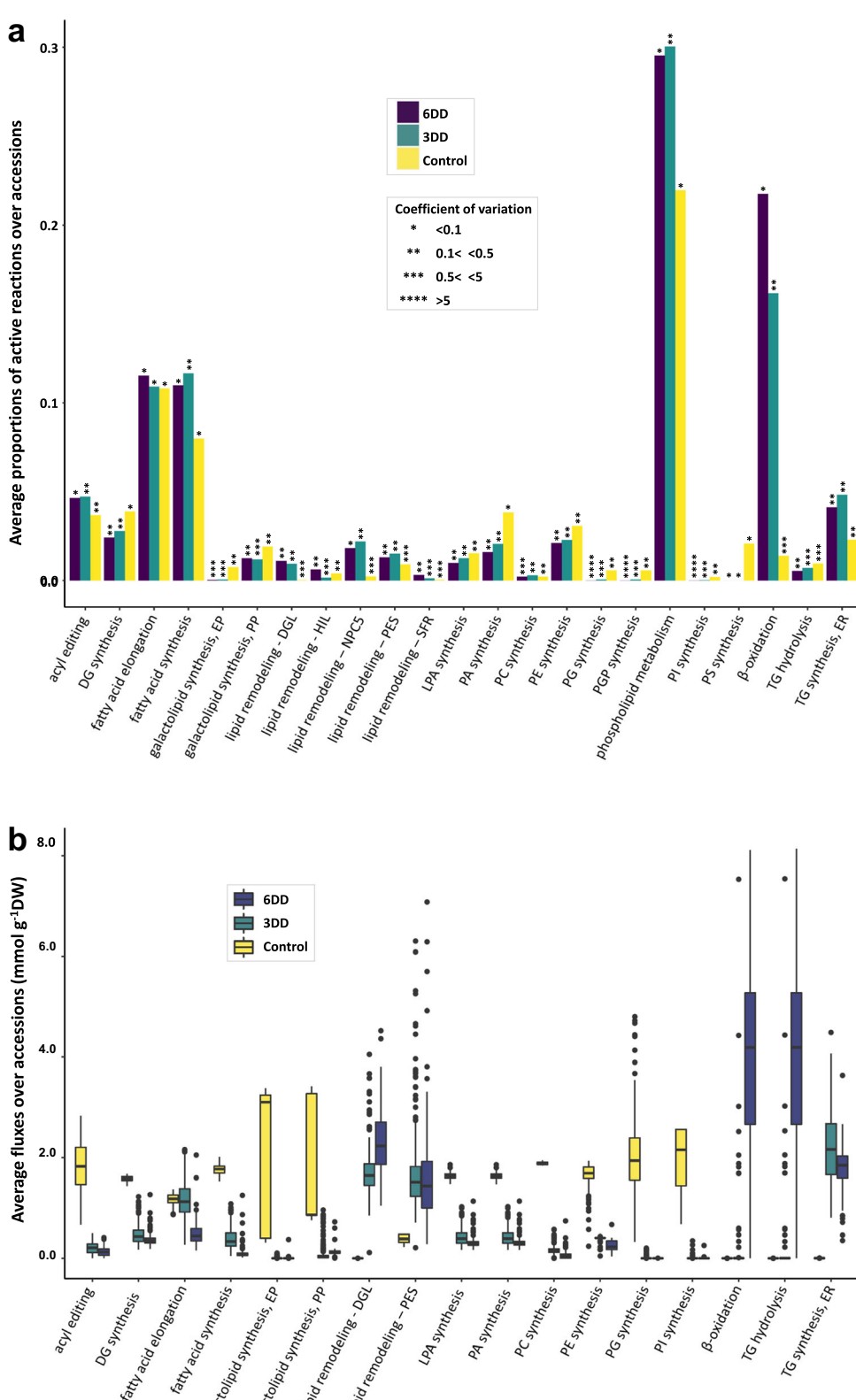

series of discrepancies, such as the identity of the enzymatic steps that catalyze specific lipid remodeling events taking place under extended darkness. These discrepancies open possibilities to use the PLM to guide future efforts in filling knowledge gaps about lipid metabolism in vegetative plant tissues. Finally, the identification of candidate genes modulating leaf lipid metabolism via flux genome-wide association yielded a valuable resource of candidate genes for

experimental validation. Of note, our findings confirmed up to 40% of regulators of lipid metabolic pathways in vegetative tissues and pave the way for model-driven discoveries in plant lipid metabolism. Together, these advances due to the refined representation of lipid metabolism in vegetative plant tissues in the PLM can be used to speed-up the engineering and selection of lines with improved lipid traits. The latter can further benefit from implementing the

**Fig. 3 | Average of reaction fluxes and proportions of active reactions over accessions. a** The proportion of active reactions per subsystem was calculated using the flux distributions obtained via pFBA, for all of the Arabidopsis accessions evaluated under three environmental conditions, e.g. illumination (control), and under extended darkness for three (3DD) and six (6DD) days. Only the subsystems related to lipid metabolism are shown, and for each case, the proportions calculated for all accessions is plotted, including the coefficients of variation of fluxes. $n = 0$ to 117 fluxes in each metabolic system are used. **b** After the flux distributions were computed via pFBA, the average flux rates (mmol g$^{-1}$DW) per subsystem were obtained for each accession by first eliminating the reactions for which physiologically meaningless flux values were obtained, followed by the estimation of the mean for each subsystem. The average flux rates were mean centered in order to

improve the visualization of subsystems with rather small reaction rate values. The number of fluxes in each metabolic system used were $n = 560$, $n = 278$ and $n = 282$ for control, 3DD and 6DD conditions, respectively. Center line, median; box limits, 75th and 25th quartiles; whiskers, 1.5×interquartile range; points, outliers. DG diacylglycerol, DGL chloroplastic galactolipase DONGLE, DW dry weight, EP eukaryotic pathway, ER endoplasmic reticulum, HIL heat inducible lipase, LPA lysophosphatidic acid, NPC5 non-specific phospholipase C5, PA phosphatidic acid, PC phosphatidylcholine, PE phosphatidylethanolamine, PES chloroplastic phytyl ester synthase, PG phosphatidylglycerol, PGP phosphatidylglycerol phosphate, PI phosphatidylinositol, PP prokaryotic pathway, PS phosphatidylserine, SFR chloroplastic galactolipid galactosyltransferase enzyme SFR2, TG triacylglycerol. Source data are provided as a Source Data file.

### Table 1 | Concordance among flux distributions and transcriptomics data sets under extended darkness[a]

| Data set 21h-XD | % agreement sign change[b] | % outliers | $\chi^2(1, N = 472)$[c] | *p*-value |
|---|---|---|---|---|
| Data set Caldana et al.[54] | 75.00 | 25.00 | 0.046 | 0.831 |
| Data set Usadel et al.[53] | 75.85 | 24.15 | | |

[a]Source data are provided as a Source Data file.
[b]The concordance in sign change was calculated among the log$_2$ of fold change (FC) for the flux distributions and two transcriptomic data sets of Arabidopsis plants under illumination (control), and 21 h of extended darkness (21h-XD).
[c]Differences between the sets ($n = 472$) was assessed by a chi-square test of independence (two-sided).

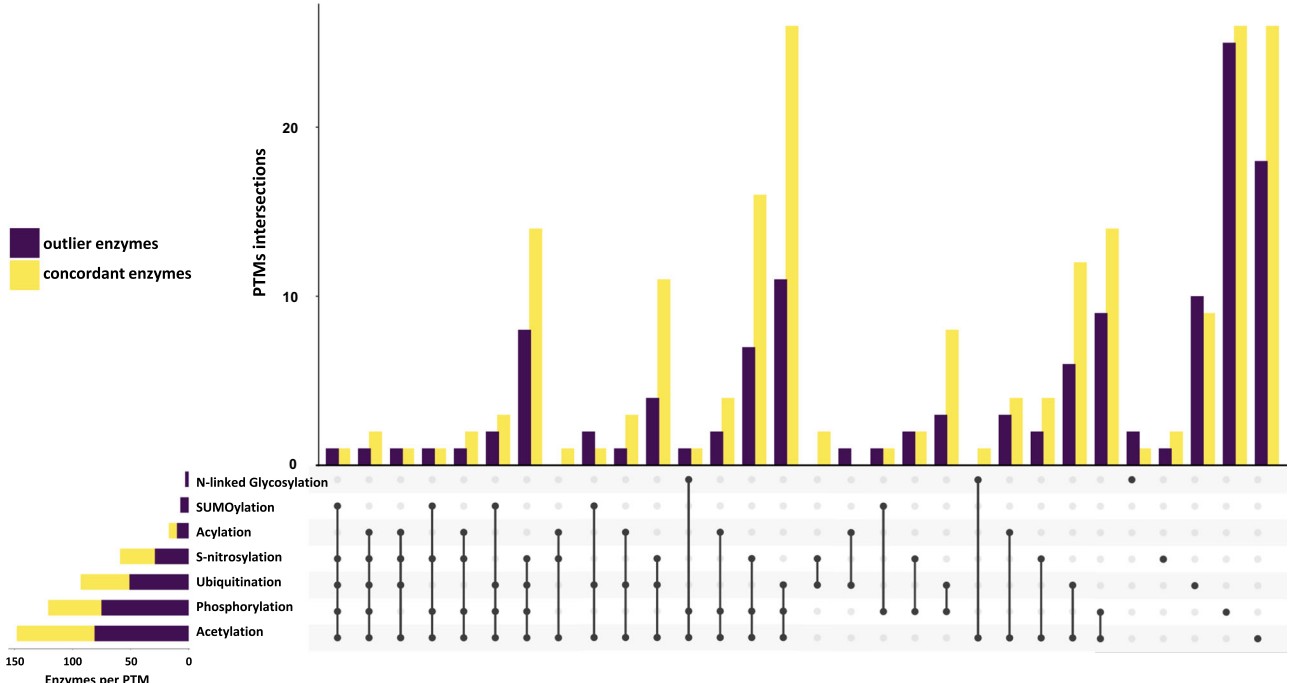

**Fig. 4 | List of post-translational modifications (PTM) in concordant and outlier enzymes.** Comparison of sign change among the log$_2$ of fold change (FC) for the flux distributions and transcriptomics data led to the classification of enzymes as concordant and outlier. The enzyme sets were further examined by listing the

post-translational modifications (PTM) identified to date. Transcriptomics data sets were obtained from Caldana et al.[54] and Usadel et al.[53] publications. Source data are provided as a Source Data file.

PLM for other species and conditions. This can be achieved by developing a pipeline to: (i) render the PLM organ-specific (e.g. seeds), by integrating organ-specific data (e.g. transcriptomics, proteomics) using well-established approaches for generating context-specific models[62]; (ii) map the orthologs that are included in the GPR rules of the PLM, making the PLM species-specific; and (iii) integrate the PLM in a condition-specific manner through the use of lipid profile measurements. These advances are expected to also improve knowledge-generation and characterization of gene functions following the proof of principle presented in this study.

## Methods

### Reconstruction of the lipid metabolic network of Arabidopsis leaves

The reconstructed PLM consists of 5956 reactions, 3108 metabolites, and 16 compartments. Since the evidence for plant lipid metabolism is incomplete, we did not rely on automated tools and performed a manual bottom-up reconstruction using all bibliomic data available for the model plant *Arabidopsis thaliana*, as well as different database resources[23,24,27,28,63–65]. The PLM includes all metabolic steps necessary for the synthesis and degradation of 26 classes of lipids

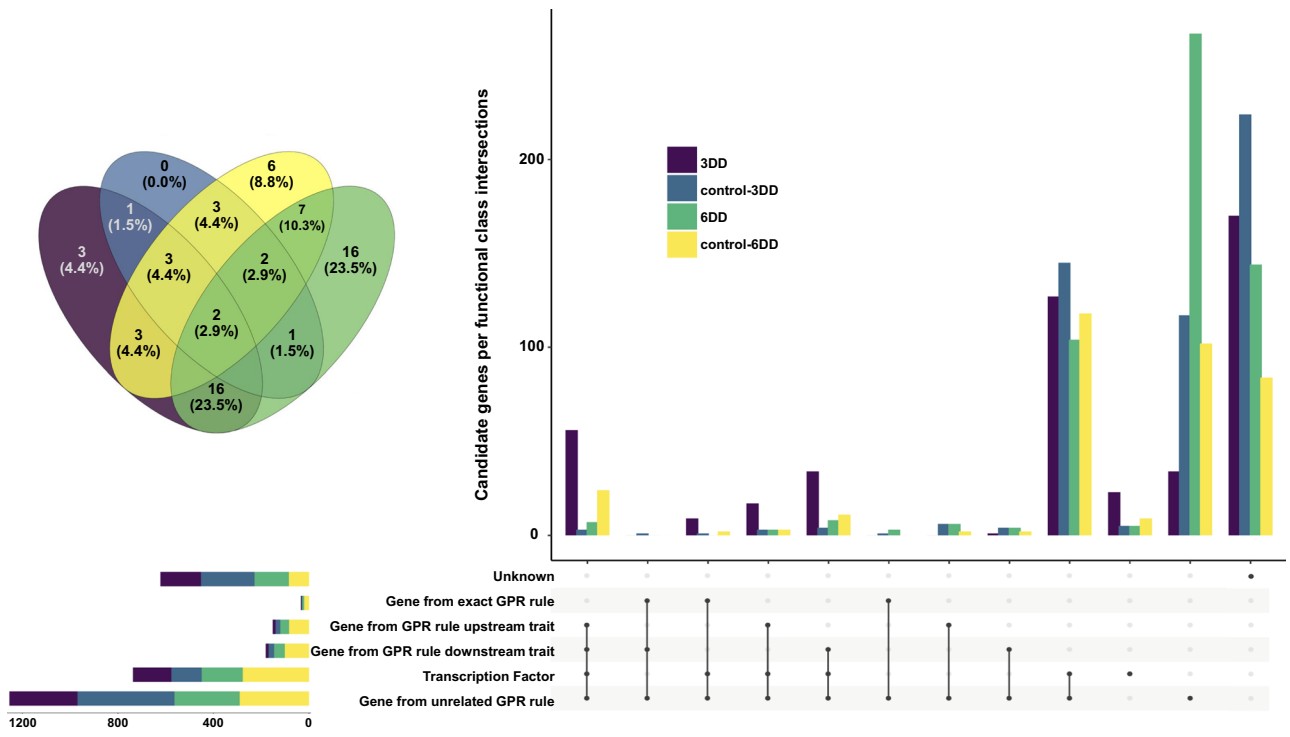

**Fig. 5 | Classification of candidate genes according to the functional role of proteins encoded.** A flux genome-wide association study allowed the identification of associated Single Nucleotide Polymorphisms for predicted fluxes under three environmental conditions, e.g. illumination (control), and under extended darkness for three (3DD) and six (6DD) days. Candidate genes were in turn classified according to the functional role of the proteins encoded by mapping them to the gene-protein-reaction (GPR) rules of the extended model, and to a database of

transcription factors (TFs)[80], hence: (i) candidates encoding TFs and candidates encoding enzymes catalyzing the reactions (ii) underlying the trait (exact GPR rule); (iii) generating precursors used by the reaction underlying the trait (GPR rule upstream); and (iv) consuming the metabolic products generated by the reaction underlying the trait (GPR rule downstream). Source data are provided as a Source Data file.

(Supplementary Data 1), comprising structural, storage, and signaling lipids, besides growth regulators and terpenoids distributed in 16 compartments, for which there is sufficient evidence regarding biosynthesis/degradation mechanisms and subcellular localization. The metabolic routes for the biosynthesis of a set of organic cofactors required for the synthesis of the lipid moieties are also included in the model. The transport reactions considered in the model are based on extensive literature search to have confidence about location, stoichiometry, and transport mechanisms (see Supplementary Data 16). The PLM also includes constraints on both lipid classes and acyl chain distributions following the SLIME approach[14], which we expanded to allow modeling of glycerolipids, glycerophospholipids, and wax monomers (see Supplementary Method 1). Demand reactions were added for several cofactors, since they were detected to participate in blocked reactions due to the coenzyme pseudo-gap problem[66]. For more details about how the reconstruction was conducted see Supplementary Method 1.

### Integration of the lipid metabolic network into existing plant models

To test the functionality of the PLM, it was initially integrated into five different published models, the AraCore model[29], the Evidenced-Arabidopsis-Model[21], the GEM created in the Path2Models project[30], the CAM diel model[31] and a medium scale model build for *Jatropha curcas*[19]. For the subsequent simulation procedures, we used the AraCore model, referred to as the "Template Model". To integrate the model, we first adjusted the Template Model to ensure physiologically relevant simulations (see Supplementary Method 2), followed by the implementation of the "LipidModuleIntegration" software in MATLAB (R2022a, The Mathworks Inc.), interfacing with COBRA Toolbox v3.0[33]

and the Gurobi Optimizer (version 8.0.1). Further details of the functionalities of the software can be found in the Supplementary Fig. 9 and Supplementary Method 2.

### Integrating quantitative information of biomass components

The biomass components were incorporated via representative metabolites or precursors: (i) Proteins: the fractional amount of the protein-bound amino acids was calculated using proteomics data under control[67] and dark[68] conditions, together with the total protein data for control[29,69] and dark-treated[70] *A. thaliana* Col-0 plants. Since the ultimate amino acids fraction in the biomass is the combination of free and protein-bound amino acids, the absolute concentration of free amino acids for control plants was obtained from[69]. For dark-treated plants, the relative abundances of soluble metabolites were measured in a population of 284 Arabidopsis natural accessions of the HapMap panel[42], then the absolute concentration of GC-MS data were obtained by multiplying the respective ratios calculated from the relative abundance data and the absolute values from control plants[69]; (ii) Starch and sugars: quantitative data for starch accumulation in dark-treated plants was obtained from[70], as hexose equivalents. The data for organic acids, sugars and several intermediates of primary and secondary metabolism was obtained by GC-MS measurements and handled as explained above for free amino acids. In brief, the absolute concentration levels of Arabidopsis (Col-0) obtained from[69] were adjusted by multiplying by the corresponding relative abundance ratios (dark- vs. light-grown plants); (iii) Cell wall and nucleic acids: to account for this biomass components, we followed the procedure described in Supplementary Method 3. Since it is not possible to obtain condition- and/or accession-specific experimental information for cell wall and nucleic acids, these components are incorporated in a

condition-unspecific manner; (iv) Lipids: since it was not possible to obtain condition- and accession-specific quantitative data for all the lipid species, a search in the literature was made to retrieve data for the absolute content of lipids in leaves of Arabidopsis plants grown under standard conditions. The values for 125 lipid species distributed in various glycero(phospho)lipid classes were obtained from[71]. Data for sphingolipid composition[27], wax precursors[72], cutin monomer composition[27,73] and total chlorophyll content[74], were obtained from the respective literature sources. In addition, a targeted lipid profiling analysis by LC–MS was performed for the previously described natural accessions. Analysis and processing of raw MS data were done with REFINER MS® 10.0 and XCalibur (Version 3.0) software. The absolute values for lipids were calculated as explained above for free amino acids using the ratios among dark- vs. light-grown plants, and the absolute content retrieved for control plants (see Supplementary Method 3 for further details).

### Creation of accession- and/or condition-specific biomass reactions

For the generation of biomass reactions, which are either condition- or accession-specific, we created scripts which retrieve the experimental data for the desired Arabidopsis accession and growth condition and calculates the stoichiometric coefficients for each biomass component (see Supplementary Method 4 for further details).

### Identification of synthetic lethal gene sets

To validate the model obtained by merging the Template Model with the Plant Lipid Module, we identified synthetic lethal sets for Arabidopsis plants cultivated under control conditions, using the Fast-SL algorithm[75]. To this end, after the integration of the Plant Lipid Module into the Template Model, we estimated the optimal relative growth rate (max $v_{bio}$) under the constraints of (i) steady-state; (ii) lower and upper flux capacities; (ii) biomass reaction under standard conditions ($v_{bio,Col0}$), (vi) the bound on the ratio between the carboxylation and oxygenation reactions catalyzed by RuBisCO was set to 2.88[76], and (v) the stoichiometric coefficients for chain- and backbone-SLIME pseudo reactions were assigned according to[14] by implementing the custom script mentioned above. Next, we constrained the model with the determined optimal relative growth and computed a reference flux distribution ($v^{Col0}$) via pFBA. We performed the synthetic lethal analysis according to[75], upon excluding demand, exchange and maintenance reactions as well as SLIME reactions. A gene was identified as essential (lethal) when biomass flux was abolished upon its removal from the model[75]. The synthetic lethal and non-lethal gene sets obtained were further examined to calculate the accuracy of the results by using information on Arabidopsis mutants available[27,34] (for further details see Supplementary Method 5 and Supplementary Data 17).

### Condition-specific flux distributions for Arabidopsis accessions

For the simulation of the extended darkness conditions we used a similar approach to[35] for the generation of a diel flux balance model where the light and dark phases are simulated simultaneously in a single optimization problem. The photoautotrophic phase was simulated by allowing a photon influx, while for the heterotrophic metabolism under dark conditions the photon influx was set to zero. We added transport reactions for a predefined list of storage compounds that are assumed to accumulate over the light conditions and are used as substrates during the dark phase. These reactions are assumed to be irreversible given that the plants were subjected to extended darkness conditions. To obtain accession-specific flux distributions, the same constraints were imposed as described in Supplementary Method 5, except for the biomass reaction and the chain- and backbone-SLIME pseudo reactions whose respective stoichiometric coefficients were calculated from measurements performed for sugars, organic acids,

amino acids, and lipids, in a population comprised of 284 Arabidopsis natural accessions belonging to the HapMap panel[42], that were subjected to extended darkness for three and six days, as explained above. Finally, the flux distribution for each accession was computed via pFBA as described in Supplementary Method 5. For more details about the construction of the diel model see Supplementary Method 6.

### Validation of the flux distribution results obtained under extended darkness

To validate the simulations under extended darkness conditions, we compared the concordance of signs between the $\log_2$–fold change (FC) values of each flux and the corresponding transcripts, measured under comparable environmental conditions. The calculation of the flux distributions via pFBA, and the construction of condition-specific biomass reactions were performed as described in Supplementary Method 6 (with the exception that soluble metabolites and lipids were obtained from Col-0 plants under control conditions and a period of 21 h of extended darkness (21h-XD)[54]). Two sets of transcriptomics data for plants under 21h-XD were obtained from previous studies[53,54], and processed first by calculating the mean of the biological replicates, and then assigning the abundance of the transcripts for each enzyme included in the model according to the GPR rules, considering the presence of isoenzymes and protein complexes[77], followed by the calculation of $\log_2$ FC (Supplementary Method 7).

### Identification of candidate genes that modulate leaf lipid metabolism

For this analysis, we made use of the flux distributions calculated for Arabidopsis accessions under extended darkness (see Condition-specific flux distributions for Arabidopsis accessions), where the fluxes were regarded as traits. For each flux, the R package rMVP (R version 4.2.1.) was used for genome-wide association studies[78]. A total of 1.329.408 SNPs from imputation of 1.001 Arabidopsis Genomes and Regional Mapping (RegMap) panel were tested, after filtering with minor allele frequency >0.05[79]. The GWAS linear mixed model considering both kinship and population structure was applied. The first three principal components were considered to represent population structure. The significant threshold was set to 1/n, where n is the number of SNPs. The significant SNPs within an interval of 20 kb were considered as one and all genes in the interval of the most significant SNPs were considered as the candidate genes.

The lists of candidates obtained were validated using information from GPR rules, and from transcription factors (TFs) and gene expression databases. Hence, from the GPR rules, the list of genes associated to each reaction flux was retrieved. Each gene was then mapped to the list of candidate genes obtained for the corresponding trait. The mapping process of the candidate genes was later extended to the GPR rules associated to the reactions immediately up- and down-stream of each of the traits for which a list of candidates was available. The next step consisted in building a list with the TFs contained in the AthaMap database[80], that was used to select the candidate genes coding for TFs. Since the list of TF-encoding candidates was substantially large, we selected only those obtained for lipid-related reactions. For each TF-encoding candidate, the list of genes with potential binding sites was obtained[80]. The latter was mapped to the GPR rules of the underlying reaction flux. We further examined the TF-encoding candidates with DNA-binding domains for the genes of the corresponding GPR rules, by reviewing their expression profiles. For this, we searched on TRAVA database[59] the expression levels of genes in whole leaves. The read counts were normalized by applying the median-of-ratio method, followed by dividing by the maximum value of expression level, so all values vary from 0 to 1[59]. The candidates were classified into four categories according to their level of expression.: (0) not expressed, (<0.3) low, (0.3 ≤ ≤ 0.6) intermediate, and (>0.6) high.

We validated the list of candidate genes by using the lipid profiles measured for a collection of 364 Arabidopsis T-DNA insertion lines[61]. To this end, the candidates were first grouped according to the type of protein they encoded (i.e. enzyme- or TF-coding gene). For the case of the enzyme-coding candidates, the metabolites associated to each reaction flux were mapped to the lipid profiles of the respective mutant, and the fold-change (mutants/wild-type) was calculated for the lipid species for which data were available (see Supplementary Method 8, Supplementary Data 14 and 18). For TF-coding candidates, we made use of the list of genes with potential DNA-binding domains obtained previously, which were mapped to the lipid profiles as in the previous case (see Supplementary Method 8, Supplementary Data 15).

### Reporting summary

Further information on research design is available in the Nature Portfolio Reporting Summary linked to this article.

## Data availability

For the reconstruction of the PLM the following databases were consulted: Aralipid website (http://aralip.plantbiology.msu.edu/)[27], the Kyoto Encyclopedia of Genes and Genomes (KEGG) (https://www.kegg.jp/), the enzyme repository BRENDA (https://www.brenda-enzymes.org/), the universal protein database (UniProt) (https://www.uniprot.org/), the Arabidopsis Information Resource (TAIR) (https://www.arabidopsis.org/), the Arabidopsis Information Portal (https://www.araport.org)[65] and the central resource for Arabidopsis protein subcellular location data (SUBA) (https://suba.live/)[23]. Source data are provided with this paper.

## Code availability

The software for the integration of the PLM is available online in Zenodo (https://doi.org/10.5281/zenodo.8179057)[81] and GitHub (https://github.com/marce2336/PlantLipidModule).

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

## Acknowledgements

The author would like to thank Dr. Camila Caldana for kindly sharing the transcriptomics and soluble metabolites data for Arabidopsis Col-0 plants under darkness. We also would like to thank A. Eckardt, A. Skirycz, A. Bolze, B. Luo, C. Scherling, D. Riewe, D. Sanchez, D. Hincha, F. Lippold, G. Wolter, J. Lisec, J. Witkowicz, L. Sieburth, L. Hong, M. Korn, M. Kraemer, M.Hundertmark, P. Do, S. Bem, S. Jozefczuk, Z. Bieniawska and Bettina Seiwert for help in harvesting Arabidopsis Col-0 plants under darkness. We are grateful to Äenne Eckardt, Gudrun Wolter and Antje Bolze for technical assistance with sample handling for UPLC/MS measurements for Arabidopsis Col-0 plants. This work was supported by the Max Planck Society and the University of Potsdam. The European Union's Horizon 2020 research and innovation program also financially supported this work through grant 862201 (to Z.N.) and project PlantaSYST (SGA-CSA No. 739582 under FPA No. 664620) (to A.R.F., S.A., and Z.N.) (this publication reflects only the author's view, and the Commission is not responsible for any use that may be made of the information it contains).

## Author contributions

Conceptualization: S.C.C. and Z.N., data curation: S.C.C., investigation: A.B., A.R.F., F.Z., H.T., S.A., and Z.N., methodology: A.B., F.Z., H.T., S.A., S.C.C. and Z.N., software: S.C.C. and Z.N. validation: S.C.C. and Z.N., writing-original draft: S.C.C. and Z.N., writing and editing: S.C.C. and Z.N. All authors reviewed and approved the manuscript.

## Funding

## Competing interests

The authors declare no competing interests.
