## [Peer Review file · Nature Communications]

Identification of gene function based on models capturing natural variability of *Arabidopsis thaliana* lipid metabolismReviewers' comments:

Reviewer #1 (Remarks to the Author):

In this manuscript, Sandra Correa and collaborators develop the Plant Lipid Module, a mechanistic model of the Plant Lipid Metabolism in Arabidopsis Rosette leaves. This model can be integrated into other metabolic models.

The authors show how the PLM can be used together with accession-specific data to generate flux balance analysis that can further be used to perform flux Genome Wide Association analysis under different conditions. The authors were able to validate up to 40% flux GWAS candidate genes using mutant lines.

This paper is a great contribution to the field. I don't have major revisions to make to the manuscript, and I think the authors present solid data to support their claims.

I think the paper would be more appealing to the wide audience of Nature Communications if the authors could be a bit more didactic in some areas of the manuscript where new concepts are introduced without much explanation. For example, the authors mentioned they used the gene-protein reaction rules to quantify the correspondence between flux and gene expression changes. The corresponding GPR paper is cited, but it would be good to explain these rules briefly.

It would also be helpful to explain how the PLM can be applied to other species and conditions, including possible limitations and necessary steps that would need to be taken.

Other than that, congratulations to Sandra Correa and co-authors.

Reviewer #2 (Remarks to the Author):

Identification of gene function based on models capturing natural variability of Arabidopsis thaliana lipid metabolism

Here the authors present a reconstructed Plant Lipid Module for Arabidopsis. This module can be readily integrated into any plant metabolic model to improve the representation of lipid metabolism within that

model. They demonstrate the module by integrating it into 5 distinct plant metabolic models where they show a capacity to improve essentiality predictions as well as predictions of lipid metabolism flux.

Overall, this work represents a very nice and comprehensive reconstruction of a specific area of metabolism, performed in an agnostic way such that the work can be easily integrated into virtually any plant modeling effort. It would be nice if more metabolic reconstruction efforts of this type were done.

I have only a few comments to the authors, mostly regarding presentation of the figures:

1.) Clearly the data underlying this model will evolve over time. Can the authors explain any plans for ongoing maintenance or updating of the module? Will there be updates to the associated module in GitHub?

2.) Did the authors run MEMOTE on their module?

3.) Were there challenges or conflicts when unifying so many disparate sources of metabolites to build the module? How were databases merged?

4.) Figure 2 is a bit challenging to understand. Maybe label (a) and (b) to clarify that they're done with the two versions of the model? Panel (c) is somewhat confusing too. We are assuming it was processed with the new module.

5.) Figure 3a it is hard to understand why the authors use stacked plots. This would seem to make the fluxes from the conditions hard to interpret. If we want to compare relative contributions to flux under different conditions, why not put bars side by side?

6.) Figure 5a - the venn diagram needs to be MUCH bigger. It's unreadable as is. In fact, all the fonts in this figure are too small. Consider breaking this up into multiple figures that can be properly sized.

Reviewer #3 (Remarks to the Author):

The authors are interested in using a comprehensive model that includes available literature to assess and then predict putative gene candidates that can alter lipid metabolism in Arabidopsis rosettes. They

have made great efforts to describe lipid metabolism and then tested the incorporation of the PLM into 5 different models to ensure functionality. Then focused on the use of the PLM with one of the models, Aracore, to perform simulations. The implementation of prediction of lethality was not completely clear or how accurate determination of lethality by the model should be stated more bluntly. Was the point that a particular critical biomass has zero flux? What is the assessment of lethality? This is not articulated clearly in the supplement where it is described around line 620. Further, if a FBA model uses biomass information to constrain fluxes, it should be no surprise that often the results will be 'correct' or a validation because the answer was provided by the data – its not an independent validation and the independent validations provided are at best expected and not great tests for the capacity of the modeling. Further the model is of a rosette, but important lipid production of Arabidopsis takes place in the seed as it is an oilseed. Other specific critiques are provided.

There are a lot of assumptions in the model with aspects taken from the literature where literature could be found, or something assumed when not found. For me this is one of the more concerning aspects about this model as well as genome scale type models that seem not well-grounded in data.

S400-500: Starch data was taken from the literature from appropriate studies; however, protein assessment contains more assumptions. Protein biomass from proteome data at 4-5 wk old Arabidopsis leaves was used – but a leaf is not really making these proteins when it is expanded as much as it is making and exporting sucrose and starch, further any making of these proteins would be partially offset by the turnover of proteins that would supply an unknown amount of amino acids and reduce the biosynthetic flux to protein. The authors discuss the proteomics data for several time points during plant development here; but it is unclear if protein is made consistently in day and night in leaves? Also, are they assuming it is made by a specific time in development? Leaves will not continue to make the same amount of protein in later development as they will in early development and in early development leaves will not be able to photosynthesize at the levels of later development. Its not clear from the literature that all of these questions are finely resolved and so it would be difficult for any modeling effort to approach this subject in a way that would give any assurance of accuracy or that it was believable.

Cell wall and lipid information were based on what studies could be found in the literature, for example a report of lipid levels in 4 week old plants. However, in the literature, usually the situation is not what the authors are considering here. Lipids are produced at different levels diurnally, however many are in a stable form and when leaves are no longer expanding in cell number then they represent semi-stable pools for which their size or amount would not matter in an FBA type model. If they or nucleotide information is being treated as though these building blocks were being made exhaustively in a four week old plant then this is inaccurate and not the way a leaf functions. Further there is not good evidence for rates of production of some of the biomass polymers so the implies that the model will be based more heavily on assumptions.

S460 Accessions were grown and subject to extended darkness, and soluble pools were measured; though it does not seem all the other biomass pools of protein, lipid etc were measured at this time. Measurements of the levels of these pools that are intermediate nodes (i.e. solubles) within metabolism seems less informative for FBA that is a steady state-based evaluation and does not assess kinetics.

Is it interesting that there is concordance between the transcript and flux? This would suggest we can disregard biochemical level regulation for which can have significant impact on lipid metabolism? Later in the manuscript it is indicated they are concordant if the 'direction is the same'. The idea is that if flux and transcript data both differ in the same way from the control then it is an concordant description. Shouldn't that be expected for an analysis where you provide biomass generation information and thus constrain fluxes? Wouldn't you expect that flux and expression data should match because the normal operation hardwired into your models will be the consumption of carbon dioxide to produce sucrose that is exported to make biomass or possibly starch?

It is unclear how the model described here overcomes the challenge of constraint-based modeling mentioned on line 55, that experimental data is sufficient to model metabolism compartmentally and overcome the 'lumping' of reactions that could consist of metabolites in different spatial locations, or for example lipids that were grouped by class but are actually different lipids for which there is less published information. It would likely be difficult to reconcile compartmental origins and fluxes for molecules shared in multiple locations, without good information – which the literature does not provide. Similarly it would be tough to model lumped lipids or metabolites as a class and say anything more specific about the individuals within that group.

It's hard to believe there is sufficient data to uniquely model 5956 reactions for 3108 metabolites across 16 compartments, or that your models would result in flux solutions that are unique.

Style:

Grammar seems fine.

It is difficult from the main text to understand what was actually done and what was actually achieved at times. As with any publication, the main document must be suitably explanatory to stand alone for the important aspects, but that seems not possible within a Nature format.

The supplements provide an organized comprehensive summary of detailed biochemical data; however, have titles like... "419589_0_data_set_7557569_rrnqcn.xlsx" – how is this meaningful? How would a reader understand which supplement to go to, to understand or inspect anything?

Response Letter to the Reviewers' comments

For completeness, all original reviews are shown in regular font and our point-by-point responses appear in bold font.

Reviewer #1:

In this manuscript, Sandra Correa and collaborators develop the Plant Lipid Module, a mechanistic model of the Plant Lipid Metabolism in Arabidopsis Rosette leaves. This model can be integrated into other metabolic models.

The authors show how the PLM can be used together with accession-specific data to generate flux balance analysis that can further be used to perform flux Genome Wide Association analysis under different conditions. The authors were able to validate up to 40% flux GWAS candidate genes using mutant lines.

This paper is a great contribution to the field. I don't have major revisions to make to the manuscript, and I think the authors present solid data to support their claims.

We thank the reviewer for the positive comments about the value of the paper and its contribution to the field.

I think the paper would be more appealing to the wide audience of Nature Communications if the authors could be a bit more didactic in some areas of the manuscript where new concepts are introduced without much explanation. For example, the authors mentioned they used the gene-protein reaction rules to quantify the correspondence between flux and gene expression changes. The corresponding GPR paper is cited, but it would be good to explain these rules briefly.

We have updated the main text by providing further information on the GPR rules and their usage for integration of transcriptomics data (see lines 188 – 192).

It would also be helpful to explain how the PLM can be applied to other species and conditions, including possible limitations and necessary steps that would need to be taken.

We have extended the discussion section to indicate how this suggestion can be carried out in future studies (see lines 287 – 295).

Other than that, congratulations to Sandra Correa and co-authors.

Reviewer #2:

Identification of gene function based on models capturing natural variability of *Arabidopsis thaliana* lipid metabolism

Here the authors present a reconstructed Plant Lipid Module for *Arabidopsis*. This module can be readily integrated into any plant metabolic model to improve the representation of lipid metabolism within that model. They demonstrate the module by integrating it into 5 distinct plant metabolic models where they show a capacity to improve essentiality predictions as well as predictions of lipid metabolism flux.

Overall, this work represents a very nice and comprehensive reconstruction of a specific area of metabolism, performed in an agnostic way such that the work can be easily integrated into virtually any plant modeling effort. It would be nice if more metabolic reconstruction efforts of this type were done.

We thank the reviewer #2 for the positive comments about the value of the paper and its contribution to the field.

I have only a few comments to the authors, mostly regarding presentation of the figures:
1.) Clearly the data underlying this model will evolve over time. Can the authors explain any plans for ongoing maintenance or updating of the module? Will there be updates to the associated module in GitHub?

For updating the module and the accompanying software tool, we plan to construct a pipeline as described in the main text (see lines 287 – 295). To this end, we have already initiated several collaborative projects that seek to integrate the PLM in other species (e.g. resurrection species, such as those from the Xerophyta genus, as well as Tomato and Potato). The new software features are planned to be made available via GitHub and Zenodo. Regarding the module's maintenance in the medium term, we plan to carry out this task within the framework of a collaborative project that seeks to make use of the PLM for the study of different aspects of lipid metabolism in *Arabidopsis*. To this end, we will monitor periodically databases and publications to identify changes in the annotation of genes related to lipid metabolism.

2.) Did the authors run MEMOTE on their module?

Memote cannot be applied with a metabolic subnetwork. However, once integrated in a model, we performed a check-up for detecting missing metabolic functions (gaps), metabolic dead-ends, mass and charge imbalances and blocked reactions. We also provide detailed annotation to the included metabolites and reactions, further contributing to easy maintenance of the PLM.

3.) Were there challenges or conflicts when unifying so many disparate sources of metabolites to build the module? How were databases merged?

One of the reasons why we carried out the reconstruction of the PLM manually was due to the multiplicity of sources necessary to obtain the required information (see Supplementary Data 2, lines 34 – 41). To unify such heterogeneous information, we contrasted all sources manually, to ensure that there was no ambiguity regarding the mechanism, stoichiometry, and subcellular localization of the reactions. In doing so, we found cases in which there were discrepancies in the

information, and to resolve these we consulted additional literature sources, whose corresponding details were included in the list of reactions of the PLM (see column 'References'). In particular cases, a description of the justification for the inclusion of particular reactions was also described (i.e. see Supplementary Data 2, lines 50 – 76; 131 – 139). It is also important to highlight that in the PLM, we did not include reactions for which it was not possible to find appropriate support. Therefore, the presented PLM is thoroughly curated and includes all non-conflicting state-of-the-art of plant lipid metabolism.

4.) Figure 2 is a bit challenging to understand. Maybe label (a) and (b) to clarify that they're done with the two versions of the model? Panel (c) is somewhat confusing too. We are assuming it was processed with the new module.

We have provided further clarification in the main text (lines 97, 100-103, 105-108) and the caption of the figure (see lines 715 – 726).

5.) Figure 3a it is hard to understand why the authors use stacked plots. This would seem to make the fluxes from the conditions hard to interpret. If we want to compare relative contributions to flux under different conditions, why not put bars side by side?

We adjusted the Figure 3a, as suggested.

6.) Figure 5a - the venn diagram needs to be MUCH bigger. It's unreadable as is. In fact, all the fonts in this figure are too small. Consider breaking this up into multiple figures that can be properly sized.

Figure 5 was modified following the suggestion of the reviewer. Figure 5b was moved to the Supplementary Data 10. The former panel (a) of Figure 5 was altered to increase the size of the Venn diagram and the size of the employed font.

Reviewer #3 (Remarks to the Author):

The authors are interested in using a comprehensive model that includes available literature to assess and then predict putative gene candidates that can alter lipid metabolism in Arabidopsis rosettes.

We have not used a comprehensive model, but we have built a comprehensive plant lipid module that was previously not available – as detailed in the introduction of the original manuscript (see lines 57 - 70). In addition, the main point of the manuscript was not to predict putative candidate genes that can alter lipid metabolism, but to provide a resource that can be readily integrated in any plant metabolic network, as stated on lines 59–60 in the original and updated version of the manuscript. We would also like to emphasize that the identification of putative candidate genes was not the main focus of our paper but was instead one of the case studies used to test the functionality of the PLM.

They have made great efforts to describe lipid metabolism and then tested the incorporation of the PLM into 5 different models to ensure functionality. Then focused on the use of the PLM with one of the models, Aracore, to perform simulations.

We thank the reviewer for recognizing this effort, which is one major contribution demonstrating the usefulness of the PLM.

The implementation of prediction of lethality was not completely clear or how accurate determination of lethality by the model should be stated more bluntly. Was the point that a particular critical biomass has zero flux? What is the assessment of lethality? This is not articulated clearly in the supplement where it is described around line 620.

It is obvious that there is a grave misunderstanding of what we have done in our analysis – which is fully detailed and reproducible.

First, we would like to clarify that the PLM *per se* does not predict the lethality of the genes. Instead, the PLM upon its integration into a metabolic model was used as the input of an algorithm for prediction of gene lethality.

Second, the identification of single and double lethal genes is well-established in constraint-based modelling (1–3). Removal of genes that cause flux through the biomass reaction to be effectively zero are considered lethal. To detect such genes we implemented a well-established algorithm, the Fast-SL (4), that allowed us to predict the single- and double-synthetic lethal genes, as was described in lines 361 – 363 of the main text, and lines 599 – 634 of the Supplementary Data 2. The Fast-SL algorithm (4) speeds up the identification of lethal reaction / genes, and is widely used (see cover letter for citation statistics). Like other approaches that address this question, Fast-SL identifies a reaction / gene as essential (lethal) when biomass flux is abolished upon its blocking / removal from the model (i.e. fixing the flux to zero). It is important to emphasize that this analysis is performed considering the presence of isoenzymes and enzyme complexes.

Third, we would like to point out that we do not understand where the confusion stems from in these standard analyses, which are widely used in the field. Nevertheless, we have adjusted the description of this analysis in the main text (lines 372 – 374) and the Supplementary Data 2 (see lines 617 – 634). In the latter, we inserted additional details to make clearer to the reader the fundamentals of this analysis and to make it even more explicit.

Further, if a FBA model uses biomass information to constrain fluxes, it should be no surprise that often the results will be 'correct' or a validation because the answer was provided by the data – its not an independent validation and the independent validations provided are at best expected and not great tests for the capacity of the modeling.

First, we do not fix the flux through the biomass reaction in any of our analyses of mutants, so the results could not be “provided by the data”, as stated by the reviewer. Hence, this cannot be an issue that affects the findings related lethality prediction.

Second, there is no other way to create accession-specific models, other than to parameterize accession-specific biomass reactions. We and others have done this line of work in many instances in modelling plant networks (5) and all pan-genome models presented to date (6–8). Hence, we do not understand what the reviewers concern is with this, also standard, procedure that is widely used in the constraint-based modelling field, as the cited studies clearly point out.

Third, even if fluxes are constrained by the biomass reactions, as the reviewer states, the mapping of fluxes to particular genes using fGWAS does not have information about the underlying genes resulting from these analyses. Therefore, the validation provided are fully independent, in contrast to the claim by the reviewer.

Further the model is of a rosette, but important lipid production of Arabidopsis takes place in the seed as it is an oilseed. Other specific critiques are provided.

We agree about the fact that important lipid production takes places in the seed of oilseed plants. However, lipid production also takes place in other plant organs, including leaves. Although lipid production in leaves is not economically relevant, its biological significance cannot be overlooked. Abundant information can be found in the literature about the roles fulfilled by lipids (9–11) that go beyond their function as an energy reservoir during seed germination. For example, we highlight the variation of traits in plant lipid metabolism in response to environmental changes (e.g. temperature), which are more substantial in vegetative tissues (12, 13) than in seeds (14, 15). This is indicative of the adaptive value of adjusting lipid composition according to environmental conditions, resulting in an optimal phenotype for a particular environment.

That said, the underestimation of the relevance of lipid metabolism in structures other than seeds has been one of the reasons why comparatively little is known about this process in leaves. It is for this reason that the PLM offers the opportunity to generate hypotheses and study how lipid metabolism contributes to the adaptation of plants to environmental conditions, which is relevant in light of climate change. In addition, the PLM can be integrated in a seed model in a future work, but this is not part of the present study and constitutes the plan that we have in the medium and long term to expand the scope and functionality of PLM. We added a paragraph on this regard in the discussion section (see lines 287 – 295).

There are a lot of assumptions in the model with aspects taken from the literature where literature could be found, or something assumed when not found. For me this is one of the more concerning aspects about this model as well as genome scale type models that seem not well-grounded in data.

First, we would like to clarify that we carried out the reconstruction of the PLM making use of the information available to date in the literature (see Supplementary Data 2 – lines 34 - 41). We also emphasize that we did not include reactions for which there was lack of evidence. Therefore, each of the metabolic pathways included in the PLM were carefully curated and are fully justified. We

therefore consider that raising a concern regarding assumptions made during the reconstruction of the PLM and that cast doubt on its quality is unfounded, particularly since no precise comments are provided in this generic comment by the reviewer's team.

Furthermore, regarding the modelling work that was carried out using the PLM, we would like to note that when using any model, constraint-based or not, it is usual to make assumptions and then validate or invalidate them in several modelling scenarios – which is precisely what we aimed to show in our study. Does the reviewer mean to say that we can only attempt to model a system / process once we know everything about the system? This is a futile endeavour, since we see modelling as a tool for hypothesis generation and testing. That said, in any publication related to modelling using metabolic models, one will find different types of assumptions, which is standard practice, since there are aspects of metabolism that are unknown even for the best characterized organisms (e.g. *Escherichia coli*, as the first organism to which constraint-based modelling was applied).

S400-500: Starch data was taken from the literature from appropriate studies; however, protein assessment contains more assumptions. Protein biomass from proteome data at 4-5 wk old Arabidopsis leaves was used – but a leaf is not really making these proteins when it is expanded as much as it is making and exporting sucrose and starch, further any making of these proteins would be partially offset by the turnover of proteins that would supply an unknown amount of amino acids and reduce the biosynthetic flux to protein. The authors discuss the proteomics data for several time points during plant development here; but it is unclear if protein is made consistently in day and night in leaves?

We must clarify that in the methodology described in Supplementary Data 2, we first described the criteria used to search for proteomics data in the literature (lines 418 – 422). Then, we specified that the proteomics data for plants cultivated under standard conditions were taken exclusively from the publication by Mergner et al. (13). In this publication, 22 days-old plants (~3- week-old) were used, and the proteomes were measured in all the rosette leaves (juveniles and adults) (lines 423 – 426), whose results were averaged since our model is not specific to a particular developmental stage. Subsequently, we described that the proteomics data for plants subjected to dark conditions were taken from the publication by Wang et al. (14), in which 3-week-old plants were also analysed (lines 433 – 438). Due to the above, it should be clear that the modelling was not carried out for different time points. Moreover, from the provided details it is also clear that the proteomics data used were obtained from studies in which the plants were cultivated under similar conditions, and whose sampling was carried out when they were in a similar stage of development (3 weeks).

In addition, throughout the manuscript we made clear that the PLM is a module created to be integrated into constraint-based (COBRA) models (line 81 of the main text). The parameterization of the model was therefore carried out according to widely used, standard procedures, which consist in integrating different data types measured in the organism of interest under specific environmental conditions. One of the approaches to integrate such data is through the inclusion of the biomass reaction, in which the stoichiometry of all molecular cell constituents is specified. For the construction of the biomass reactions, we did an exhaustive search for experimental data measured in *Arabidopsis thaliana* plants that were subjected to environmental conditions similar to those that were simulated. Additionally, we measured several parameters (e.g. lipids, soluble metabolites), which were also used for the construction of the biomass reactions. This guarantees that the modelling was carried out with realistic data on the composition of the leaf biomass under the environmental conditions that were simulated. In the simulations, a pseudo-steady state of

metabolism was assumed, and as a result the predict flux distribution provides a snapshot of reaction fluxes in one stage of development. Generation of time-resolved fluxes would require further assumptions, which are difficult to justify without additional measurements. Therefore, we consider the comment by the reviewer unrealistic, particularly considering the fact that such dynamic flux balance analysis models are available only for single-cell organisms with simpler developmental processes than those existing in plants.

That said, in the case of the simulation under extended dark conditions, we generated a diel flux balance model (see main text lines 378 – 385), in which the light and dark phases were simulated simultaneously in a single optimization problem. This is a well-established approach that is also widely used in the plant constraint-based modelling (see the cover letter for the citation statistics). Hence, for parameterization of the diel model, we used experimental data measured in plants under light and dark conditions, to simulate the characteristic behaviour of leaves under the photoautotrophic and heterotrophic phases, respectively.

We are convinced that these details fully address the points raised by the reviewer's team.

Also, are they assuming it is made by a specific time in development? Leaves will not continue to make the same amount of protein in later development as they will in early development and in early development leaves will not be able to photosynthesize at the levels of later development. Its not clear from the literature that all of these questions are finely resolved and so it would be difficult for any modeling effort to approach this subject in a way that would give any assurance of accuracy or that it was believable.

As specified in the answer to the previous comment, we assume quasi-steady state in a given developmental stage (~3 weeks) for which we obtained or re-used data for parameterization of the biomass reactions. We agree that modelling full plant development would necessitate further assumptions, as stated in our answer above. For this reason, we focus on analysing representative flux distributions for 3-week-old rosette.

Cell wall and lipid information were based on what studies could be found in the literature, for example a report of lipid levels in 4 week old plants. However, in the literature, usually the situation is not what the authors are considering here. Lipids are produced at different levels diurnally, however many are in a stable form and when leaves are no longer expanding in cell number then they represent semi-stable pools for which their size or amount would not matter in an FBA type model. If they or nucleotide information is being treated as though these building blocks were being made exhaustively in a four week old plant then this is inaccurate and not the way a leaf functions. Further there is not good evidence for rates of production of some of the biomass polymers so the implies that the model will be based more heavily on assumptions.

As explained above, we did not carry out simulations of different developmental stages, and therefore, we made sure that all available data used was obtained from plants grown under similar conditions and of particular developmental stage. In addition, the PLM does not incorporate information about the rate of formation of any pools over diurnal cycle, since the model is based on quasi-steady state assumption which is implicit to constraint-based modelling. Hence, this concern of the reviewer is fully addressed. Finally, we wish to emphasize that the components of the biomass reactions were not arbitrarily included. The inclusion of each component was made based on our measurements, as well as experimental data from publications in which the plants were grown under similar environmental conditions.

S460 Accessions were grown and subject to extended darkness, and soluble pools were measured; though it does not seem all the other biomass pools of protein, lipid etc were measured at this time. Measurements of the levels of these pools that are intermediate nodes (i.e. solubles) within metabolism seems less informative for FBA that is a steady state-based evaluation and does not assess kinetics.

First, lipids were also measured under the extended darkness condition (see Supplementary Data 2, lines 523 – 532 and 541 – 558). Second, we did not build a dynamic model -- all other models (16), including the diel extensions, are based on steady-state assumptions, therefore the concern about the lack of a kinetic assessment is not well-grounded.

Is it interesting that there is concordance between the transcript and flux? This would suggest we can disregard biochemical level regulation for which can have significant impact on lipid metabolism?

We start by saying that lipid metabolism is transcriptionally regulated, hence one would hypothesize good agreement between the responses of transcript levels and fluxes to changes in environment. This is the hypothesis that we have tested, and we clearly stated this in the manuscript. We do not follow where the " suggestions" that the reviewer mentions come from, especially since we follow our analysis with additional tests that consider post-transcriptional mechanisms (see main text lines 199 – 208), which surprisingly neglected the mention by the reviewer.

Later in the manuscript it is indicated they are concordant if the 'direction is the same'. The idea is that if flux and transcript data both differ in the same way from the control then it is an concordant description. Shouldn't that be expected for an analysis where you provide biomass generation information and thus constrain fluxes?

No, this is not readily expected. Predictions of fluxes do not rely on integration of transcriptomics data. If what the reviewer says is always true, then transcript changes would be an excellent prediction for biomass change – which we know does not hold in practice (see for example one of the data-driven studies in this direction (17)). Therefore, we do not see the logic used in founding this concern.

Wouldn't you expect that flux and expression data should match because the normal operation hardwired into your models will be the consumption of carbon dioxide to produce sucrose that is exported to make biomass or possibly starch?

No, fluxes are predicted without relying on transcriptomics data and hence no match is readily expected. This is a hypothesis which we explicitly test, please, see the comment above for clarification. We do not follow the claim about hardwiring of metabolism, as this contradicts the claim for the lack of unique flux distributions that the reviewer brings up in the last major comment, below.

It is unclear how the model described here overcomes the challenge of constraint-based modeling mentioned on line 55, that experimental data is sufficient to model metabolism compartmentally and overcome the 'lumping' of reactions that could consist of metabolites in different spatial locations, or for example lipids that were grouped by class but are actually different lipids for which there is less published information. It would likely be difficult to reconcile compartmental origins and fluxes for

molecules shared in multiple locations, without good information – which the literature does not provide. Similarly it would be tough to model lumped lipids or metabolites as a class and say anything more specific about the individuals within that group.

The reviewer fully missed the fact that the PLM provides a fine-grained description of lipid metabolism and precisely overcomes the lumping. In fact, we appreciate it that the reviewer mentions that it would be challenging to model lumped lipids and that it would be difficult to reconcile compartmental origin -- but this is exactly what the PLM allows us to do. Thus, several key features were considered and included in the construction of the PLM: (i) None of the pathways incorporated in the PLM have lumped reactions; (ii) Each of the lipid species and their corresponding intermediates have complete information at the structural level, as well as the respective identification of the subcellular location in which they were generated (see Supplementary Data 2, lines 265 - 277); (iii) The only lipid species that were grouped corresponded to some sn-positional isomers that share the same origin (prokaryotic or eukaryotic route) and subcellular location (the latter was emphasized in the Supplementary Data 2) (see Supplementary Data 2, lines 269 - 273), which helps to reduce the computational burden without losing the traceability of each species. Here, we would like to emphasize that such grouping was not performed for lipid classes, but in rather specific cases; and (iv) In the modelling procedure for lipid metabolism we also introduced the approach described by Sanchez et al. (18) (see Supplementary Data 2, lines 204 – 222), which allows a more realistic simulation of lipid metabolism (see Supplementary Data 2, lines 223 – 227). Additionally, the features (ii) - (iv) allow the use of lipidomics data generated with low or high resolution lipidomics platforms for the imposition of constraints on the fluxes of lipid-related reactions.

Its hard to believe there is sufficient data to uniquely model 5956 reactions for 3108 metabolites across 16 compartments, or that your models would result in flux solutions that are unique.

In deriving steady-state flux distributions, we rely on parsimonious Flux Balance Analysis (pFBA) – a standard tool that has been widely used in analysis of genome-scale (i.e. large) plant and other metabolic networks (see the cover letter for the number of citations). pFBA substantially reduces the solution space, particularly when used in a two-step procedure as described in our Supplementary Data 2 (see lines 601 – 624). Therefore, this concern is also fully addressed. Nevertheless, to provide further evidence, fully dispelling the concerns, we performed a flux variability analysis (FVA) with parsimonious flux balance analysis (pFBA). The FVA was conducted at optimum biomass (constraining the biomass reaction with the maximum-predicted growth rate) and minimizing the flux through gene associated reactions by holding minimum network flux constant. The latter was done by first minimizing the absolute value of flux through all gene-associated reactions via pFBA, and next using this flux to constrain the upper bound for the summed network flux. Upon comparing the flux distributions (see file 'results_FVA.xlsx'), we found that the difference among the minimum and maximum flux values for 79% of the reactions that carry flux is close to zero ($< 0.35 \text{ mmol g}^{-1}\text{DW}$). The remaining reactions (21%) show a difference that ranges from $3 \text{ mmol g}^{-1}\text{DW}$ up to the maximum value set for the upper bound (1000), from which 65% correspond to reversible reactions, which is fully justified. In the calculated FVA flux distributions, there is a group of reactions not carrying flux. For 60% of these reactions, the inactivity is condition-specific, which means that they are used in the simulation of lipid remodelling events taking place under different stress conditions. The remaining 40% inactive reactions are producing different lipid species and their intermediates which were not included in the biomass reactions, due to the absence of quantitative information for the condition simulated. Updating the biomass reaction in different scenarios (e.g. species, organs) will allow further activation of these reactions.

Style:

Grammar seems fine.

It is difficult from the main text to understand what was actually done and what was actually achieved at times. As with any publication, the main document must be suitably explanatory to stand alone for the important aspects, but that seems not possible within a Nature format.

We cannot give a concrete answer to this comment since the reviewer is not pointing to any specific section of the manuscript that the reviewer's team considers unclear. All subsection titles summarize the presented findings in that section and can be easily linked to the methods section. We would appreciate it if the reviewer's team clearly states what exactly should be adjusted to improve the readability of the text.

The supplements provide an organized comprehensive summary of detailed biochemical data; however, have titles like... "419589_0_data_set_7557569_rrnqcn.xlsx" – how is this meaningful? How would a reader understand which supplement to go to, to understand or inspect anything?

When we submitted the manuscript, all the supplementary files were prepared following the instructions from the journal, by naming each file as 'Supplementary Data x', where the x stands for the consecutive number that was used. The title that the reviewer includes as example does not correspond to any of the names used in our supplementary files. This must be due to how the journal handles the submitted files internally or to any other error during the processing of the files. We cannot provide further details at this point since we cannot see where the issue is. To support our argument, we attach the report of the supplementary files submitted where the name of the original files can be clearly seen (see file 'FileDescriptions.pdf').

List of references cited:

1. O. Güell, F. Sagués, M. Á. Serrano, Essential Plasticity and Redundancy of Metabolism Unveiled by Synthetic Lethality Analysis. *PLoS Comput. Biol.* **10**, e1003637 (2014).
2. P. F. Suthers, A. Zomorodi, C. D. Maranas, Genome-scale gene/reaction essentiality and synthetic lethality analysis. *Mol. Syst. Biol.* **5**, 301 (2009).
3. R. Harrison, B. Papp, C. Pál, S. G. Oliver, D. Delneri, Plasticity of genetic interactions in metabolic networks of yeast. *Proc. Natl. Acad. Sci. U. S. A.* **104**, 2307–2312 (2007).
4. A. Pratapa, S. Balachandran, K. Raman, Fast-SL: an efficient algorithm to identify synthetic lethal sets in metabolic networks. *Bioinformatics* **31**, 3299–3305 (2015).
5. H. Tong, A. Küken, Z. Nikoloski, Integrating molecular markers into metabolic models improves genomic selection for Arabidopsis growth. *Nat. Commun.* **11**, 2410 (2020).
6. H. Lu, *et al.*, A consensus *S. cerevisiae* metabolic model Yeast8 and its ecosystem for comprehensively probing cellular metabolism. *Nat. Commun.* **10**, 1–13 (2019).
7. K. Correia, R. Mahadevan, Pan-Genome-Scale Network Reconstruction: Harnessing Phylogenomics Increases the Quantity and Quality of Metabolic Models. *Biotechnol. J.* **15**, 1900519 (2020).
8. H. Tong, A. Küken, Z. Razaghi-Moghadam, Z. Nikoloski, Characterization of effects of genetic variants via genome-scale metabolic modelling. *Cell. Mol. Life Sci.* **78**, 5123 (2021).

9. W. Dowhan, M. Bogdanov, E. Mileykovskaya, "CHAPTER 1 - Functional roles of lipids in membranes" in *Biochemistry of Lipids, Lipoproteins and Membranes (Fifth Edition)*, D. E. Vance, J. E. Vance, Eds. (Elsevier, 2008), pp. 1–37.
10. D. K. Allen, P. D. Bates, H. Tjellström, Tracking the metabolic pulse of plant lipid production with isotopic labeling and flux analyses: Past, present and future. *Prog. Lipid Res.* **58**, 97–120 (2015).
11. N. N. Li, C. C. Xu, Y. H. Li-Beisson, K. Philippar, Fatty Acid and Lipid Transport in Plant Cells. *Trends Plant Sci.* **21**, 145–158 (2016).
12. A. Burgos, *et al.*, Analysis of short-term changes in the Arabidopsis thaliana glycerolipidome in response to temperature and light. *Plant J.* **66**, 656–668 (2011).
13. Y. Higashi, Y. Okazaki, F. Myouga, K. Shinozaki, K. Saito, Landscape of the lipidome and transcriptome under heat stress in Arabidopsis thaliana. *Sci. Rep.* **5**, 1–11 (2015).
14. A. L. Vuorinen, *et al.*, Effect of growth environment on the gene expression and lipids related to triacylglycerol biosynthesis in sea buckthorn (*Hippophae rhamnoides*) berries. *Food Res. Int.* **77**, 608–619 (2015).
15. C. R. Linder, Adaptive Evolution of Seed Oils in Plants: Accounting for the Biogeographic Distribution of Saturated and Unsaturated Fatty Acids in Seed Oils. <https://doi.org/10.1086/303399> **156**, 442–458 (2000).
16. C. Y. Maurice Cheung, M. G. Poolman, D. A. Fell, R. George Ratcliffe, L. J. Sweetlove, A Diel Flux Balance Model Captures Interactions between Light and Dark Metabolism during Day-Night Cycles in C3 and Crassulacean Acid Metabolism Leaves. *Plant Physiol.* **165**, 917–929 (2014).
17. M. Westhues, *et al.*, Omics-based hybrid prediction in maize. *Theor. Appl. Genet.* **130**, 1927–1939 (2017).
18. B. J. Sanchez, F. Li, E. J. Kerkhoven, J. Nielsen, SLIMEr: probing flexibility of lipid metabolism in yeast with an improved constraint-based modeling framework. *Bmc Syst. Biol.* **13**, 4 (2019).

Reviewers' Comments Rebuttal:

Reviewer #1 (Remarks to the Author):

The authors have addressed all my comments.

Reviewer #2 (Remarks to the Author):

The authors responded well to all my claims and other claims in general.

Reviewer #6 (Remarks to the Author):

I was brought in as additional reviewer of this revised manuscript as reviewer 3 was no longer available. I briefly reviewed the MS, the rebuttal letter, and some of the supplements in particular the methods supplement. It is clear to me that this has been a massive undertaking leading to an important contribution to the field. Having contributed some of the key enzyme and gene identifications through the reductionist approach myself, I believe that the described modeling approach based on the existing literature data and the provided additional data are necessary to provide a comprehensive picture of plant lipid metabolism and its integration into overall cellular metabolism to move the field forward. While I did not take a deep dive into the actual model and algorithms behind it, I am familiar with the general principles and think that the demonstrated integration and testing of the model is well done and meets current standards of analysis. The high-level description in the body of the article is adequate and improved in the current version as requested by the reviewers and the supplemental data are extensive and fully supporting the conclusions. I also find that the discussion of how the model was produced and what was included (supplement two) is expertly done, comprehensive and respectful towards existing literature, and accurately reflects the state of current knowledge. Obviously, the underlying gene annotations will need to be revised as hardcore biochemists will continue to provide new data on individual enzyme and gene functions, and it is clear that the reductionist approach and the described modeling approaches have to go hand in hand to move the field forward. I agree with reviewer 2 that the model needs constant updating as new data becomes available and the authors' response seems adequate in this regard. The bottom line for me as a plant lipid biochemist is that the described model was long overdue and will move the field forward and I thank the authors for their effort and contribution to the field.